# Increasing Prevalence of Occult HBV Infection in Adults Vaccinated Against Hepatitis B at Birth

**DOI:** 10.3390/vaccines13020174

**Published:** 2025-02-12

**Authors:** Ge Zhong, Zhi-Hua Jiang, Xue-Yan Wang, Qin-Yan Chen, Lu-Juan Zhang, Li-Ping Hu, Mei-Lin Huang, Yu-Bi Huang, Xue Hu, Wei-Wei Zhang, Tim J. Harrison, Zhong-Liao Fang

**Affiliations:** 1Guangxi Zhuang Autonomous Region Center for Disease Prevention and Control, Guangxi Key Laboratory for the Prevention and Control of Viral Hepatitis, Nanning 530028, China; gxzhongge@163.com (G.Z.); jiangzhihuagxcdc@163.com (Z.-H.J.); yanziwang007@hotmail.com (X.-Y.W.); chenqingyan060@sina.com (Q.-Y.C.); zhanglujuan777@163.com (L.-J.Z.); wfhlp2007@163.com (L.-P.H.); ml-huang@outlook.com (M.-L.H.); huangyubi0603@163.com (Y.-B.H.); hx2010708966@163.com (X.H.); zhangww6717@163.com (W.-W.Z.); 2School of Preclinical Medicine, Guangxi Medical University, Nanning 530021, China; 3Division of Medicine, University College London Medical School, London WC1E 6BT, UK; t.harrison@ucl.ac.uk

**Keywords:** hepatitis B virus (HBV), occult HBV infection, hepatitis B vaccine, vaccination, prevalence

## Abstract

Background/Objectives: Immunization with the hepatitis B vaccine is the most effective means of preventing acute HBV infection. However, whether the primary vaccination of infants confers lifelong immunity remains controversial. Therefore, the ongoing surveillance of vaccine recipients is required. Methods: A longitudinal study was carried out based on LongAn county, one of the five clinical trial centers for hepatitis B immunization in China in the 1980s. Serum samples were collected and tested for HBV serological markers and DNA. Results: A total of 637 subjects born in 1987–1993 were recruited, including 503 males and 134 females. The total prevalence of HBsAg was 3.9%. The prevalence in females (8.2%) was significantly higher than that in males (2.8%) (*p* = 0.004). The prevalence of anti-HBc in females (52.2%) was also significantly higher than that in males (41.2%) (*p* = 0.021). The prevalence of anti-HBs was 42.7% and did not differ significantly between males (41.7%) and females (46.3%) (*p* = 0.347). Compared to data from surveillance over the last ten years, the positivity rate of HBsAg did not increase. The positivity rate of anti-HBs decreased significantly (*p* = 0.049) while that of anti-HBc increased significantly (*p* = 0.001). The prevalence of occult HBV infection (OBI) in 2024 (6.0%) was significantly higher than that in 2017 (1.6%) (*p* = 0.045). Subjects diagnosed with OBI in 2017 maintained occult infection in 2024. Conclusions: Neonatal HBV vaccination maintained effective protection for at least 37 years. However, the prevalence of OBI increases with age in those vaccinated at birth, raising a new issue of how to prevent and control OBI in the post-universal infant vaccination era.

## 1. Introduction

Despite global immunization programs, persistent infection with hepatitis B virus (HBV) remains a major public health problem [1]. The infection, which may lead to acute and chronic liver diseases, including cirrhosis and hepatocellular carcinoma (HCC) [1], is transmitted perinatally, horizontally between children in infancy, and via the percutaneous route [2]. More than two billion people, one-third of the world’s population alive today, have been infected with HBV and approximately 257 million remain persistently infected [3]. Regarding HBV-related HCC, it is estimated that the number of associated deaths was 192,000 in 2019, an increase from 156,000 in 2010 [4]. 

The first hepatitis B vaccine was licensed in 1981 [5] and immunization has become the most effective means of preventing acute infection by HBV [6]. In 1992, the World Health Organization set a goal for all countries to introduce the hepatitis B vaccine into the national Expanded Program on Immunization (EPI) by 1997 [7]. Globally, the prevalence of chronic HBV infection among children below 5 years of age fell from 4.7% to 1.3% over two decades [8]. A remarkable reduction in HBV-related diseases was observed in Italy in the first three decades of universal vaccination [9]. The universal vaccination of newborns reduced the incidence of HCC in Asia, compared to the pre-vaccination era [10]. However, whether immunization of neonates provides lifelong protection remains contentious, because immunity to HBV may deteriorate over time [11,12]. Individuals with high initial levels of HBV immunity may experience a faster rate of decline of antibody titers [13]. Therefore, the ongoing surveillance of vaccine recipients is needed to determine whether primary hepatitis B vaccination can confer lengthy, or even lifelong, protection [6].

Meanwhile, more and more evidence reveals that occult HBV infection (OBI), which is defined as the presence of HBV DNA in the liver or serum of individuals who test negative for HBsAg using currently available assays, may occur in individuals vaccinated at birth [14,15]. These individuals may be those born to HBsAg-negative mothers [16] or those born to mothers who were positive for HBsAg, despite hepatitis B immunoglobulin being administered at birth [17]. OBI may become overt after several years [18]. It has been reported that OBI is associated with the development of HCC [19,20].

Prior to the advent of immunization against HBV, around 17.2% of the adult population LongAn, a county in Guangxi, China, were carriers of HBsAg [21]. LongAn hosts one of the five clinical trials of hepatitis B immunization in China and, between 1986 and 1996, all newborn infants in these trials were vaccinated according to a 0-, 1-, and 6-month schedule using a 10 μg dose of plasma-derived HBV vaccine, regardless of the mother’s HBV infection status [22]. Since 1997, all newborn infants in LongAn county have been vaccinated with the recombinant HBV vaccine. Our surveillance ten years ago revealed that the HBsAg and anti-HBc positivity rates in LongAn county had increased, 22–28 years after immunization. Anti-HBs positivity and the geometric mean concentration of antibodies decreased significantly in the years after vaccination [23]. We also found later that a few individuals vaccinated at birth had become persistently infected with HBV [24]. Has the prevalence of HBsAg in this vaccinated cohort increased since then? Does OBI prevail in this vaccinated cohort? Here, we report our surveillance data.

## 2. Materials and Methods

### 2.1. Study Population

As mentioned above, LongAn hosts one of the five clinical trials of hepatitis B immunization in China. In order to evaluate the efficacy of vaccination, a serial study has been carried out in LongAn county since the early 1990s. In the past ten years, there have been four visits, in 2015, 2017, 2019, and 2020. The candidate study subjects for this study were those who were born between 1987 and 1993 and vaccinated at birth with three doses of HepB vaccine and who came to our evaluation at least once in four visits in the past ten years and provided serum samples (Figure 1). In 2024, doctors from local town hospitals and village clinics dispensed notice to all candidate study subjects. The subjects who were willing to attend our visit became be our study subjects. Each study subject completed a one-page questionnaire at that visit and provided a 3 mL sample of blood by venepuncture for testing for serological markers of HBV and HBV DNA. The questionnaire included demographic information, including sex, birth date, ethnicity, place of birth, and immunization history. History of diseases such as autoimmune liver disease, metabolic liver disease, generalized metabolic disorders, etc., and blood transfusion history were recorded.

The inclusion criteria included the following: ① born between 1987 and 1993; ② came to our evaluation at least once; ③ received three 10 μg doses of a plasma-derived HB vaccine at the ages of 0, 1, and 6 months; ④ no history of blood transfusion; ⑤ no autoimmune liver disease, metabolic liver disease, generalized metabolic disorders, etc. Those who had received the first dose of the vaccine more than 72 h after birth were excluded.

Informed consent in writing was obtained from each individual. The study protocol conformed to the ethical guidelines of the 1975 Helsinki Declaration and was approved by the Guangxi Institutional Review Board.

### 2.2. Qualitative Assays of HBV Serological Markers

Sera were tested for HBV serological markers (including hepatitis B surface antigen (HBsAg), hepatitis B surface antibody (anti-HBs), and hepatitis B e antigen (HBeAg)), antibodies to the hepatitis B e antigen (anti-HBe), and antibodies to the hepatitis B core antigen (anti-HBc), using enzyme immunoassays (WANTAI BioPharm, Beijing, China). Quality control for the measurements was performed in accordance with the protocols provided by the manufacturer.

### 2.3. Quantitative Assays of Anti-HBs and HBsAg

The serum anti-HBs and HBsAg concentrations were quantified by the Maccura i6000, using HBsAg and HBsAb Quantitative Kits (Maccura Biotechnology Co., Ltd., Chendu, China). The Maccura i6000 is a fully automated chemiluminescence immunoassay (CLIA) system designed for high-throughput testing in clinical laboratories. According to protocols provided by the manufacturer, positive and negative cutoffs were calculated with the positive and negative controls, as required by the diagnostic kits. The dynamic range of the kit for HBsAg was 0.05~250 IU/mL. The dynamic range of the kit for anti-HBs was 4~1000 mIU/mL.

### 2.4. Measurement of Serum Viral Loads

Serum HBV DNA concentrations were quantified by real-time polymerase chain reaction (PCR) using commercial reagents (Sansure Biotech Inc., Changsha, China) in an ABI Prism 7500 sequence detection system (Applied Biosystems, Foster City, CA, USA), with a dynamic range of 30–5 × 10^9^ IU/mL. The viral loads of serum samples collected in 2017 were also measured for comparison. To avoid the effect of cross-contamination on the results, negative and blank controls were included in each assay. Positive test results were repeated three times.

To confirm OBI, HBV genomic DNA was extracted from 200 μL of serum samples using QIAamp DNA Mini kits (QIAGEN GmbH, Hilden, Germany) and eluted in 50 μL of distilled water. To amplify the pre-S, polymerase, and X regions, a first round of PCR was carried out in a 50 μL reaction using the primers LSOB1 (nt 2739–2762, 5′-GGCATTATTTGCATACCCTTTGG-3′) and MDN5R (nt1794–1774, 5′-ATTTATGCCTACAGCCTCCT-3′) with a 5 min hot start followed by 30 cycles of 94 °C for 30 s, 50 °C for 30 s, and 72 °C for 90 s. A second round of PCR was carried out with 5 μL of the first round’s products in a 50 μL reaction using the primers LSBI1 (nt 2809–2829, 5′-TTGTGGGTC ACCATATTCTT-3′) and XSEQ1R (nt1547–1569, 5′-CAGATGAGAAGGCACAGACGGGG-3′), with the same amplification protocol as the first round. Products from the second round were confirmed by agarose gel electrophoresis.

### 2.5. Statistical Methods

Categorical data were evaluated by χ^2^ or Fisher’s exact tests, depending on the absolute numbers included in the analysis, and quantitative data were analyzed by the independent sample *t* test or Mann–Whitney U test. HBV DNA concentrations were set at 30 IU/mL for those with undetectable values. All *p* values were 2-tailed, and *p* < 0.05 was considered significant. All statistical analyses were performed using SPSS software (ver. 16.0; Chicago, IL, USA).

## 3. Results 

### 3.1. General Characteristics of Study Subjects

The 637 subjects in this study were recruited from the LongAn cohort, including 503 males and 134 females, which indicated a response rate of 20.5% (637/3108) among the candidate study subjects. The average age was 34.0 ± 2.0 (Mean ± SD) years. The average ages of the males and females were 34.0 ± 2.0 and 34.0 ± 1.9, respectively. The total prevalence of HBsAg was 3.9% ((95% CI (confidence interval): 2.4–5.4)). The prevalence of HBsAg in females (8.2%) was significantly higher than that in males (2.8%) (*p* = 0.004). The total prevalence of anti-HBc was 43.5% (95% CI: 39.7–47.3). The prevalence of anti-HBc in females (52.2%) was also significantly higher than that in males (41.2%) (*p* = 0.021). The total prevalence of anti-HBs was 41.7% (95% CI: 37.9–45.5). However, there was no significant difference in its prevalence between males (41.7%) and females (46.3%) (*p* = 0.347). Clearly, the prevalence of HBsAg corresponded to the prevalence of anti-HBc (Table 1).

### 3.2. Trends in the Prevalence of HBsAg and the Positive Rates and Levels of Anti-HBs, According to Age

The highest prevalence of HBsAg in males (3.6%) was seen in the age group of 36, while that in females (18.8%) was in the age group of 30. The prevalence of HBsAg varied with age, and it can be seen that the prevalence of HBsAg increased with age in males but not in females (Figure 2). 

The highest prevalence of anti-HBs positivity in males (52.5%) was seen in the age group of 36, while that in females (68.8%) was in the age group of 30. The prevalence of anti-HBs positivity varied with age and it can also be seen that the prevalence of anti-HBs positivity increased with age in males but not in females. The levels of anti-HBs corresponded to the positive rates of anti-HBs in males but not in females (Figure 3). These data suggest that using the positive rate or titer of anti-HBs to determine the protection period of HepB vaccination may not be helpful.

### 3.3. Trend in Prevalence of Anti-HBc According to Age

The highest prevalence of anti-HBc in males (51.8%) was seen in the age group of 36, while that in females (68.8%) was in the age group of 30. The prevalence of anti-HBc varied with age. However, no trend was seen in either males or females (Figure 4).

### 3.4. Characteristics of Subjects Positive for HBeAg

We found 17 subjects to be positive for HBeAg but negative for HBsAg. Eleven of them were positive for anti-HBs. In order to determine whether these subjects had OBI, we tested for HBV DNA using qPCR. It was surprising that only one of them was positive for HBV DNA, with a low viral load of 80 IU/mL. Clearly, only one of these subjects had OBI (Table 2).

### 3.5. Comparison of Prevalence of HBsAg, Anti-HBs and Anti-HBc from Same Subject, Previously and Currently (2024)

We performed four tests in 2015, 2017, 2019, 2020, and 2024, respectively. The comparison of serological markers was performed between the recent tests and the test in 2024. Among the 637 subjects, the HBsAg status (positive or negative) of 98.9% (95%CI: 98.1–99.7) of subjects remained unchanged. One subject alone became positive for HBsAg, while six subjects underwent the seroconversion of HBsAg from positive to negative. The difference between the rate of becoming positive for HBsAg and the rate of the seroconversion of HBsAg from positive to negative was not significant (*p* = 0.070). It was noted that one of the six subjects was positive for HBsAg in 2009, 2017, and 2019 and another was positive for HBsAg in 2017, 2019, and 2020. Clearly, the positivity rate of HBsAg did not increased (Table 3).

Most of the subjects (90.4%, 95% CI: 88.1–92.7) maintained their anti-HBs status. The percentage of those who became negative for anti-HBs (5.5%) was significantly higher than that of those who became positive for anti-HBs (4.1%) (*p* = 0.049), suggesting that the rate of anti-HBs positivity decreased with age (Table 3).

The anti-HBc status of 71.3% (67.8–74.8) of the 637 subjects remained unchanged. However, the percentage of subjects who became anti-HBc positive (24.6%) was significantly higher than that of those who became anti-HBc negative (4.1%) (*p* = 0.001), suggesting that the positivity rate of anti-HBc increased with age (Table 3).

### 3.6. Comparison of Occult Infection Between Subjects Recruited in 2017 and 2024

We tested for HBV DNA in 282 subjects in 2024 and 129 subjects in 2017, respectively. The total prevalences of OBI in 2017 and 2024 were 1.6% (95% CI: −0.6–3.8) and 6.0% (3.2–8.8), respectively. The difference in the prevalence of OBI between 2017 and 2024 was significant (*p* = 0.045), suggesting that the prevalence of OBI in those vaccinated at birth increased with age.

It is noted that 82.4% (95% CI: 67.8–100.6) of OBI occurred in those who were anti-HBc positive; more than half of them (9/16) were positive for anti-HBs and some had high levels of anti-HBs, suggesting that anti-HBs may not prevent occult HBV (Table 4 and Table 5). 

Viral loads in the subjects with OBI were low, except for one subject whose viral load was 4.1 × 10^4^ IU/mL. The viral loads of 89.5% (17/19) of subjects were less than 10^3^ IU/mL.

## 4. Discussion 

To our knowledge, this is the first study comparing the long-term efficacy of vaccination against hepatitis B in the same individuals at different time points. The major findings were that the prevalence of OBI increased with age. The rate of anti-HBc positivity increased with age and the rate of anti-HBs positivity decreased with age. The positivity rate of HBsAg did not increased. The prevalence of anti-HBc positivity corresponded to that of both HBsAg and OBI. Using the positive rate or titer of anti-HBs to determine the protection period of HepB vaccination may not helpful. Anti-HBs may not prevent occult HBV. A strength of this study is that comparing the long-term efficacy of vaccination in the same person at different time points may have avoided sampling bias and yielded reliable results. A weakness of the study is that there were fewer females than males because we failed to contact them, which may have prevented stratification analysis. According to custom in LongAn county, a woman must move to live with her husband in other villages or towns or counties, which resulted in us losing contact details. 

There have been many studies evaluating the long-term efficacy of HepB vaccination. They have found that the titer or levels of anti-HBs decrease with ages [25,26,27]. The longest period of protection with HepB vaccination is 35 years [28]. In this study, we found that the prevalence of HBsAg did not increase significantly, compared to that in 2015, but the prevalence of anti-HBc doubled [15,29], suggesting that the protection period of HepB vaccination was at least 37 years.

It has been reported that, despite HepB vaccination and passive protection with hepatitis B immunoglobulin (HBIG), OBI is common in children and adolescents in high-risk groups [12,26]. This may occur in anti-HBs-positive young adults after neonatal HepB vaccination [30]. It is more common in those born to HBsAg-positive mothers [31,32]. However, these were all cross-sectional studies and could not determine whether the prevalence of occult infection increased or decreased. In this study, we are the first to show that the prevalence of OBI in those vaccinated at birth increased significantly, compared to data from the same subjects in 2017. 

The factors involved in the development of OBI, despite vaccination and prophylaxis with HBIG, remain obscure. One potential factor is the presence of mutations in the region of the HBV genome encoding the surface protein; these may lead to failure to detect HBsAg and apparent HBsAg negativity [33]. However, we reported previously that five of six subjects with OBI did not have unusual amino acid substitutions in or around the major antigenic region of HBsAg (the “a” determinant) [34]. Pollicino et al. did not find any relevant pre-S deletion mutations in their occult HBV isolates and found that these isolates could replicate normally and produce HBsAg. They proposed that host, rather than viral, factors are responsible for OBI [35]. Under immune suppression, HBV can still produce very small amounts of antigens that are not detectible by available assays. Co-infection with HCV or HIV can also contribute to the appearance of OBI by interfering either with the host immune system or with the replication of HBV [36]. Another potential host factor is the gut microbiota, which may contribute to the immune response, leading to suppressed virus replication and the development of occult hepatitis B [37]. 

Occult HBV can be transmitted and develop as an overt infection [18,38]. It has been implicated in the development of chronic liver diseases, including cirrhosis and HCC [20,39]. It may also reactivate in the face of immunosuppression, even leading to the development of fulminant hepatitis [40]. We previously reported that the prevalence of OBI in those without vaccination in LongAn county was 11.5% [34]. Although the prevalence of OBI among adults vaccinated against hepatitis B at birth is lower than 11.5%, it increases with age, raising a new issue of how to prevent and control OBI in the post-universal infant vaccination era.

Anti-HBc develops during acute HBV infection and may be maintained for an entire life. Thus, the serological status of anti-HBc alone is compatible with resolved acute infection, as well as overt or occult chronic HBV infection [41]. Studies from high-prevalence areas confirm the potential for waning immunity to allow breakthrough infections to occur. Although chronic HBV infections after vaccination are rare, the prevalence of anti-HBc has been seen to increase with time [8]. In this study, we found that OBI occurred most frequently in subjects who were anti-HBc positive, especially those with anti-HBc alone. We also found that the prevalence of anti-HBc is increasing. Therefore, the risk of occult HBV should no longer be neglected in vaccinated populations.

Clinically, HBeAg is an indicator of viral replication and infectivity, inflammation and the severity of disease, and the response to antiviral therapy. Seroconversion from HBeAg-positivity to an HBeAg-negative or anti-HBeAg-positive phase usually heralds the resolution of infection [42]. Although, in general, HBeAg positivity correlates with a high serum level of HBV DNA, it has been reported that low levels of HBeAg and HBV DNA are not always in proportion, but very high levels of HBeAg are often correlated with high levels of HBV DNA [43]. In this study, 16 subjects who were HBsAg-negative but positive for HBeAg were found to be negative for HBV DNA. Were the levels of HBV DNA in these subjects below the limit of detection? An overall 97–100% rate of anti-HBs development at ≥10 mIU/mL could be achieved in those with a full course of primary HepB vaccination [6]. Over time, anti-HBs concentrations decline in vaccinated individuals, becoming undetectable in some [8]. However, vaccinees with anti-HBs <10 mIU/mL or undetectable anti-HBs still have a brisk anamnestic immune response to HBsAg [5,8]. In this study, we found that 4.1% of subjects were negative for anti-HBs in 2017 but positive in 2024. These subjects denied having received a booster. We postulate that they may have been exposed to wild-type virus. Therefore, using the titer of anti-HBs to determine the protection period of HepB vaccination may not helpful.

In this study, we found that two subjects with OBI in 2017 maintained occult infection in 2024. This is not in correspondence with the data from Lai’s group [44]. We also found that 15 subjects were negative for HBV DNA in 2017 but had developed OBI by 2024. Because we did not test their family members, such as their spouses, parents, brothers, and sisters, for HBV, the sources of their infections remained unclear. Furthermore, we did not determine the sequences of the S gene of the virus and could not determine whether mutants were responsible for their occult infection.

Our data show that the prevalence of HBsAg increases with age in males but not in females. This may be because males have more social interaction than females of an equivalent age, exposing them to infectious sources of HBV. In this study, we found that OBI occurred most frequently in subjects who were anti-HBc positive, especially those with anti-HBc alone. It has been reported that OBI is associated with the development of HCC [16,17]. Future investigation is therefore warranted to determine whether there is an association between anti-HBc positivity and important health consequences such as liver cancer and HBV reactivation with immune suppression.

The major limitation of our study is that we did not test family members for HBV, the results of which may have helped us determine whether the infection originated from the family or beyond. If it was from the general population, this may help explain the increasing prevalence of OBI, with age, in adults vaccinated against HBV, because they have more social interaction with increasing age. Another limitation is that we did not determine the sequences of the S gene of the virus, which should help to identify the cause of OBI. In the future, we will search for infectious sources and determine the sequences of the S gene of HBV.

In conclusion, the prevalence of OBI increased with age in individuals vaccinated against hepatitis B at birth. The prevalence of anti-HBc positivity corresponded to that of OBI. Neonatal HBV vaccination maintained effective protection for at least 37 years in terms of the positivity rate of HBsAg and anti-HBs. 

## Figures and Tables

**Figure 1 vaccines-13-00174-f001:**
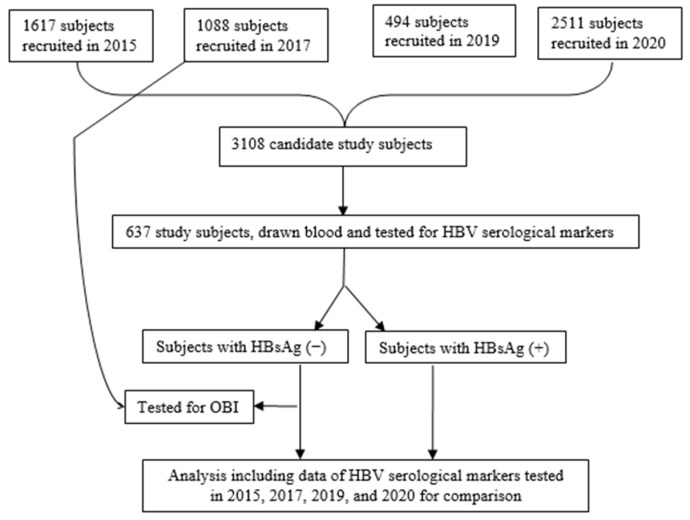
Flowchart describing study protocol.

**Figure 2 vaccines-13-00174-f002:**
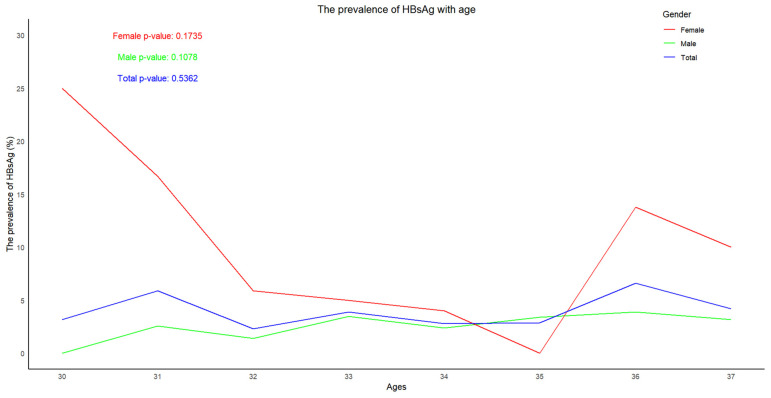
Trend in prevalence of HBsAg according to age.

**Figure 3 vaccines-13-00174-f003:**
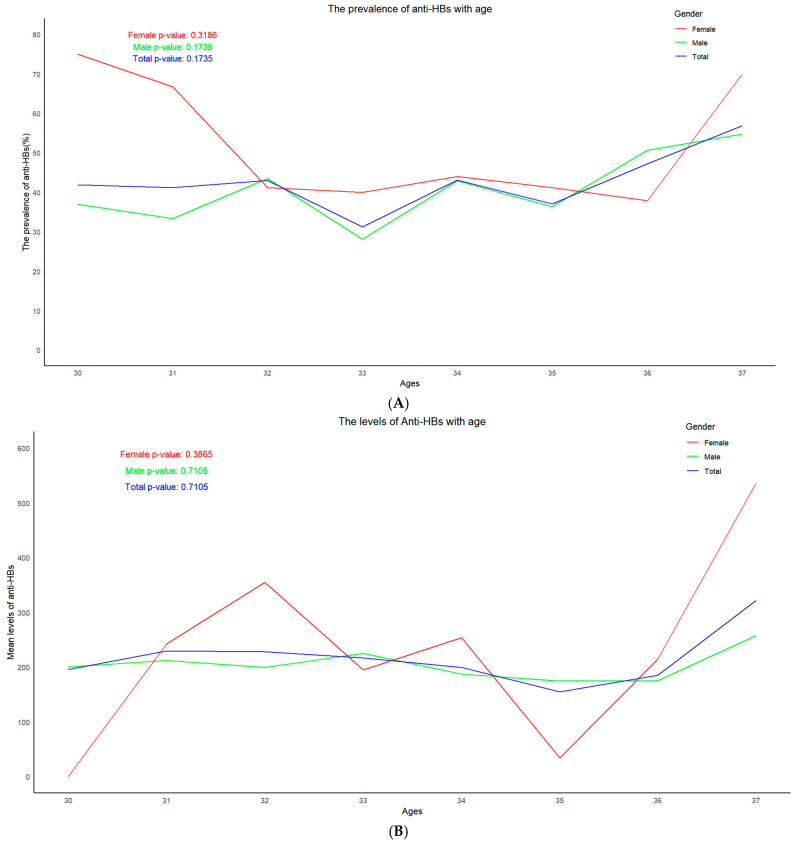
Trend in positive rates and levels of anti-HBs according to age. (**A**) Trend in positive rates according to age. (**B**) Trend in levels of anti-HBs according to age.

**Figure 4 vaccines-13-00174-f004:**
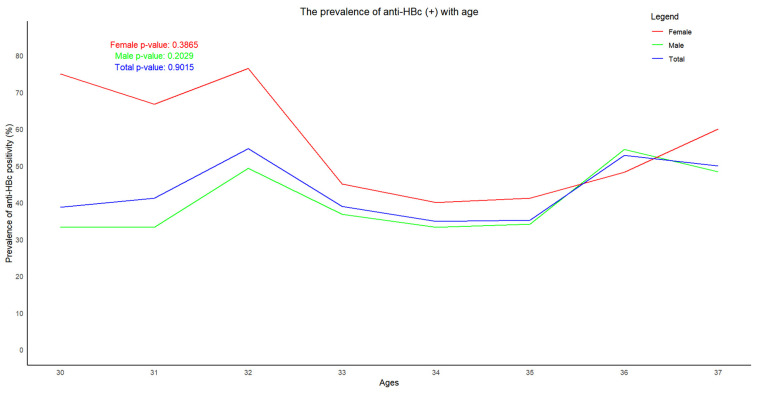
Trend in prevalence of anti-HBc according to age.

**Table 1 vaccines-13-00174-t001:** The prevalence of serological markers of HBV in 2024.

Groups	Males			Females			Total		
	No.	Positive	Prevalence (%) (95% CI ^△^)	No.	Positive	Prevalence (%) (95% CI)	No.	Positive	Prevalence (%) (95% CI)
HBsAg	503	14	2.8 (1.5–4.1)	134	11	8.2 (6.1–10.3)	637	25	3.9 (2.4–5.4)
Anti-HBs	503	210	41.7 (37.9–45.5)	134	62	46.3 (42.4–50.2)	637	272	42.7 (38.9–46.5)
Anti-HBc	503	207	41.2 (36.9–45.5)	134	70	52.2 (43.7–60.7)	637	277	43.5 (39.7–47.3)
Levels of Anti-HBs	Mean ± SD ^#^			Mean ± SD			Mean ± SD		
	200.9 ± 323.2 ^§^			266.8 ± 353.2			215.5 ± 330.4		

^#^ SD: standard deviation. ^△^ CI: confidence interval. ^§^: IU/mL.

**Table 2 vaccines-13-00174-t002:** Characteristics of subjects positive for HBeAg in 2024.

Codes	Gender	Ages	HBsAg	anti-HBs	HBeAg	anti-HBe	anti-HBc	Viral Loads
AWL120	M *	33	-	16.428	+	-	-	Undetectable
AWB133	F ^#^	32	-	-	+	-	-	Undetectable
ATZ196	M	33	-	-	+	-	-	Undetectable
ATX297	M	34	-	334.995	+	-	+	Undetectable
ATX289	M	35	-	-	+	-	-	Undetectable
ATX261	F	33	-	-	+	-	-	Undetectable
ATW230	M	34	-	190.843	+	-	-	Undetectable
ATT524	M	37	-	>1000.000 ^§^	+	-	-	Undetectable
ATM174	M	33	-	>1000.000	+	-	+	Undetectable
ATL181	M	31	-	-	+	-	-	80.009 ^△^
ATL143	M	31	-	78.288	+	-	+	Undetectable
ATF074	M	32	-	98.957	+	-	+	Undetectable
AQT182	M	35	-	>1000.000	+	-	-	Undetectable
APP045	M	35	-	-	+	-	-	Undetectable
ACS020	M	31	-	482.039	+	-	-	Undetectable
09LA1043	M	34	-	35.529	+	-	+	Undetectable
09LA0167	M	34	-	>1000.000	+	-	-	Undetectable

* Males. ^#^ Females. ^§^ IU/mL. ^△^ IU/mL.

**Table 3 vaccines-13-00174-t003:** Changes in prevalence of HBsAg, anti-HBs, and anti-HBc in individuals, up to 2024.

Marker	No.	Status (Positive or Negative) Unchanged		Became Positive		Became Negative	
		No.	Rate (%) (95% CI *)	No.	Rate (%) (95% CI)	No.	Rate (%) (95% CI)
HBsAg	637	630	98.9 (98.1–99.7)	1	0.2 (−0.1–0.5)	6	0.9 (0.1–1.6)
Anti-HBs	637	576	90.4 (88.1–92.7)	26	4.1 (2.6–5.6)	35	5.5 (3.7–7.3)
Anti-HBc	637	454	71.3 (67.8–74.8)	157	24.6 (21.3–27.9)	26	4.1 (2.6–5.6)

* CI: confidence interval.

**Table 4 vaccines-13-00174-t004:** Comparison of occult infection rates between subjects recruited in 2017 and 2024.

Groups	No.	Positive	Rate (%)	95% CI *
2024				
Negative for all serological markers	62	1	1.6	−1.5–4.7
anti-HBc (+) only	89	7	7.9	2.3–13.5
anti-HBc (+) with anti-HBs (+) or anti-Hbe (+) or HbeAg (+)	104	7	6.7	1.9–11.5
anti-HBc (−) with anti-HBs (+) or anti-Hbe (+) or HbeAg (+)	27	2	7.4	−2.5–17.3
Total	282	17	6.0	3.2–8.8
2017				
Negative for all serological markers	53	1	1.9	−1.8–5.6
anti-HBc (+) only	38	0	0	0
anti-HBs (+), anti-HBc (+)	29	1	3.4	−3.2–10.0
Total	129	2	1.6	−0.6–3.8

* CI: confidence interval.

**Table 5 vaccines-13-00174-t005:** Characteristics of subjects with occult infection.

Code	Gender	Age	HBsAg	anti-HBs	HBeAg	anti-HBe	anti-HBc	HBV DNA (IU/mL)
09LA1051	M ^△^	34	-	248.835 ^§^	-	-	+	157.965 ^§^
09LA1073	F ^#^	32	-	-	-	-	+	102.424
09LA3607	M	34	-	-	-	-	+	253.2
ADH140 *	F	33	-	-	-	-	-	41192.4
ADH140	F	33	-	-	-	-	-	267.372
ADJ006	F	36	-	243.483	-	-	+	1842.646
ADJ102	M	32	-	-	-	-	+	223.013
ADK129	M	37	-	435.870	-	+	+	49.213
ADK130	M	37	-	-	-	+	-	447.401
AGM067	F	33	-	-	-	-	+	180.14
AGM098	M	35	-	-	-	-	+	307.819
AGM212	M	36	-	337.604	-	-	+	436.139
AGZ078	M	36	-	-	-	-	+	108.223
ATJ035	M	32	-	944.213	-	-	+	263.29
ATL181	M	31	-	-	+	-	-	80.009
ATW293	M	36	-	-	-	-	+	695.963
ATW295 *	M	32	-	+	-	-	+	96.331
ATW295	M	33	-	269.656	-	-	+	212.238
ATZ238	M	37	-	595.325	-	-	+	65.842

* Samples collected in 2017. ^△^ Male. ^#^ Female. ^§^ IU/mL.

## Data Availability

The data presented in this study are available within the article.

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
