# Peer review of "Increasing Prevalence of Occult HBV Infection in Adults Vaccinated Against Hepatitis B at Birth"

_vaccines, 2025, doi:10.3390/vaccines13020174_

Round 1
Reviewer 1 Report
Comments and Suggestions for Authors
Dr Zhong and colleagues described HBsAg prevalence in China in a cohort of subjects vaccinated for HBV at birth, in a region with high prevalence of infection.
However, there are several issue.
The definition of OBI is missing in the methods.
In which year was the study conducted?
When did the patients first visit the clinic for the sample collection?
What is the follow-up time after the first sample collection?
Table 3 should define the timing between the first and second tests.
As part of a clinical study, data on how many subjects responded to the HBV vaccine at birth must be added. How many were tested for anti-HBs after the three doses of the vaccine? How many did not respond to the vaccine?
In Table 1, if they were vaccinated and responders, the number of positive for anti-HBs would be close to 100%.
How many people are vaccinated each year at that clinic? To understand whether the sample of 637 people in the study is representative of the entire population or if the individuals visiting the clinic had a clinical reason / recent non-familial HBV transmission risk factor. This is also a limitation of the study. How many arrived at enrollment with acute HBV hepatitis? Positive anti-HBc IgM? How many with altered ALT?
Subjects born to HBsAg-negative mothers cannot be OBI, but it is plausible that they contracted the infection in childhood, in a region where 19% of the population is HBsAg positive overall.
Not having tested the mothers and lacking pre-vaccine serological data, some subjects may have contracted HBV at birth and later recovered. This would explain the higher percentage of anti-HBc positive subjects in 2024 compared to 2017 (majority of mother-to-child transmission in the past compared to youngest). In this case, the conclusions of the study are incorrect.
It is an interesting study only to describe the number of OBI and subjects with previous HBV in China in 2017 and 2024, but everything else is not methodologically sound.
Author Response
The definition of OBI is missing in the methods.
Answer: Agreed. The definition of OBI has been added when it first appears in the Introduction.
In which year was the study conducted?
Answer: Agreed. We have added “2024” to the Study Population section.
When did the patients first visit the clinic for the sample collection?
What is the follow-up time after the first sample collection?
Answer: We apologise for any confusion.
As mentioned in the Study Population section, in order to evaluate the efficacy of vaccination, a serial study has been carried out in LongAn county since the early 1990s. Over the last ten years, there have been four visits, in 2015, 2017, 2019 and 2020. The subjects who attended were not the same at each visit. Therefore, one of our recruitment criteria is that candidate study subjects came for our evaluation at least once in the last ten years and provided a serum sample. We have modified this section accordingly.
Table 3 should define the timing between the first and second tests.
Answer: Agreed, this has been done. We have added “We have four tests in 2015, 2017, 2019 and 2020, respectively, in the last ten years. The comparison of serological markers was done between the most recent test and the test in 2024.” This is outlined in section 3.5.
As part of a clinical study, data on how many subjects responded to the HBV vaccine at birth must be added. How many were tested for anti-HBs after the three doses of the vaccine? How many did not respond to the vaccine?
Answer: The universal infant vaccination program was started in LongAn county in 1987. All of our study subjects were born between 1987-1993. The immunization programme did not involve testing for anti-HBs after the three doses of the vaccine. Therefore, we compared the HBV serological markers between the latest test and the test in 2024.
In Table 1, if they were vaccinated and responders, the number of positive for anti-HBs would be close to 100%.
Answer: This view is not supported by the data from other studies (Posuwan N, et al. PLoS One. 2018 Aug 20;13(8):e0202637. Su TH, et al. Emerg Microbes Infect. 2012 Sep;1(9):e27 ).
How many people are vaccinated each year at that clinic?
Answer: We do not have these data. All infants were vaccinated in local clinic and the population varied among the clinics.
To understand whether the sample of 637 people in the study is representative of the entire population or if the individuals visiting the clinic had a clinical reason / recent non-familial HBV transmission risk factor. This is also a limitation of the study.
Answer: The study subjects were recruited, based on our previous studies (Li H, et al. Epidemiol Infect. 2017 Apr;145(5):887-894). Therefore, they should be representative of the entire population. None of our study subjects have a clinical condition. However, it is possible for them to have non-familial HBV transmission risk factors.
How many arrived at enrollment with acute HBV hepatitis? Positive anti-HBc IgM? How many with altered ALT?
Answer: We carried out a physical examination of all subjects at enrollment. We did not detect acute hepatitis although we did not test for anti-HBc IgM. We investigated the ALT levels only of those with OBI, all were normal.
Subjects born to HBsAg-negative mothers cannot be OBI, but it is plausible that they contracted the infection in childhood, in a region where 19% of the population is HBsAg positive overall.
Answer: We and others reported previously similar data (Hu LP, et al. PLoS One. 2015 Oct 12;10(10):e0138552. ; Hsu, et al. Hepatology. 2015 Apr;61(4):1183-91). Hsu reported that "73% of OBI children with known maternal HBV status had carrier mothers". In another word, about 27% OBI subjects contracted the infection in childhood.
Not having tested the mothers and lacking pre-vaccine serological data, some subjects may have contracted HBV at birth and later recovered. This would explain the higher percentage of anti-HBc positive subjects in 2024 compared to 2017 (majority of mother-to-child transmission in the past compared to youngest). In this case, the conclusions of the study are incorrect.
Answer: We cannot understand the point of this comment. Our study subjects are at least thirty years old. If they had contracted HBV at birth and later recovered, anti-HBc could be positive in 2017 but have become negative in 2024. If it is negative in 2017 and positive in 2024, the infection must be recent.
It is an interesting study only to describe the number of OBI and subjects with previous HBV in China in 2017 and 2024, but everything else is not methodologically sound.
Answer: We do not agree with this comment, we have six sections of results and OBI is only one aspect.
Reviewer 2 Report
Comments and Suggestions for Authors
The authors have taken a unique cohort of longitudinally followed people after infant HBV vaccination to determine changes in HBV serological status over time. This cohort was previously reported in 2017 from samples collected in Feb 2015. In this study, samples from 2024 were compared to prior samples to see if additional participants had developed HBsAg or anti-HBc, as well as newly tested HBeAg, anti-HBe or HBV DNA.
Most public health authorities believe that effective vaccination at any age will prevent development of chronic active HBV infection. This has led to the recommendation not to revaccinate people who have documentation of a full HBV vaccine series, even if their anti-HBs levels are low/undetectable (see recent USA ACIP recommendations as an example). What is less discussed is the risk of exposure and possible integration of HBV without active infection, as seen in people who are HBsAg negative and anti-HBc positive. Presumably these people are at risk for HBV reactivation if they develop future immunosuppression.
This study has shown continued rising prevalence of anti-HBc over time. They have also shown an increase in occult HBV in these vaccinated adults. The most important consequence of this study is whether we should, in fact, be giving HBV vaccine boosters to adults with anti-HBs titers less than some IU/mL threshold.
Questions and comments:
1) What does the title, “…at birth running…” mean?
2) The text and Tables have both 2017 and 2015 as dates of prior HBV serologies. If two different dates were used this needs to be explained in Methods, and samples from each time point need to be clearly marked.
3) Line 85: “Those subjects who are willing to attend our visit became be our study subjects.” This sentence reads oddly. These individuals have been study subjects since they were enrolled at birth, right? So I think study participant respondents who consented to follow up were included in this study.
4) Line 141 (prevalence), Table 1 (rate/percent), line 190 (positivity rate), Line 198 (percentage) all seem to be describing the same types of data. In particular, “rate” implies data over time. I can’t figure out what the unit of time is. Table 1 states 2024, but the time unit cannot be one year (2024) since I am reviewing this manuscript in 2024.
5) Table 1: The prevalence/rate/percent of females with anti-HBc is 52.2, but line 141 uses the number 46.3%. I can’t tell if these are different units of measurement or this is a mistake. There are several other data points that seem discrepant between text and tables – please review all numbers carefully.
6) Line 150: Could there have been differences in late vaccination rates between males and females or other explanations for this difference?
7) Line 209: Elsewhere the authors suggest that high titers of anti-HBs may be reflective of boosting from natural infection. Do the authors have previous anti-HBs titers to determine whether these people had low titers at some point in the past?
8) Line 235: This study demonstrates that we need to be more precise in our terminology. Birth vaccination provides strong (but not complete) protection from the development of chronic, active HBV infection (HBsAg positive) but protects less well from exposure (anti-HBc). The obvious question, which perhaps could be a future objective of this study, is whether detection of anti-HBc is associated with important health consequences (liver cancer, reactivation with immune suppression, etc). If that link is established, then the world may need to reconsider revaccinating people in adulthood.
9) Line 279: Those of us who practice in low-endemicity regions rarely see unusual combinations of HBV serologies such as HBsAg negative, HBeAg positive and HBV DNA negative. This study is important in demonstrating what is possible and what serology tests we should be following.
Comments on the Quality of English Language
This manuscript needs editing to improve clarity.
Author Response
The authors have taken a unique cohort of longitudinally followed people after infant HBV vaccination to determine changes in HBV serological status over time. This cohort was previously reported in 2017 from samples collected in Feb 2015. In this study, samples from 2024 were compared to prior samples to see if additional participants had developed HBsAg or anti-HBc, as well as newly tested HBeAg, anti-HBe or HBV DNA.
Most public health authorities believe that effective vaccination at any age will prevent development of chronic active HBV infection. This has led to the recommendation not to revaccinate people who have documentation of a full HBV vaccine series, even if their anti-HBs levels are low/undetectable (see recent USA ACIP recommendations as an example). What is less discussed is the risk of exposure and possible integration of HBV without active infection, as seen in people who are HBsAg negative and anti-HBc positive. Presumably these people are at risk for HBV reactivation if they develop future immunosuppression.
This study has shown continued rising prevalence of anti-HBc over time. They have also shown an increase in occult HBV in these vaccinated adults. The most important consequence of this study is whether we should, in fact, be giving HBV vaccine boosters to adults with anti-HBs titers less than some IU/mL threshold.
Questions and comments:
1) What does the title, “…at birth running…” mean?
Answer: This seems to be a problem with the editing of the layout. The title is “Increasing prevalence of occult HBV infection in adults vaccinated against hepatitis B at birth”. The running title is “Occult HBV infection and HepB vaccination”.
2) The text and Tables have both 2017 and 2015 as dates of prior HBV serologies. If two different dates were used this needs to be explained in Methods, and samples from each time point need to be clearly marked.
Answer: we agree. We have added “Viral loads of serum samples collected in 2017 were also measured for comparison.” to the section of 2.4. Furthermore, we also corrected 2015 in Table 4 and in Discussion.
3) Line 85: “Those subjects who are willing to attend our visit became be our study subjects.” This sentence reads oddly. These individuals have been study subjects since they were enrolled at birth, right? So I think study participant respondents who consented to follow up were included in this study.
Answer: No, these individuals have been study subjects who came for our evaluation at least once in the last ten years. We have modified accordingly the section of 2.1 to make this clear.
4) Line 141 (prevalence), Table 1 (rate/percent), line 190 (positivity rate), Line 198 (percentage) all seem to be describing the same types of data. In particular, “rate” implies data over time. I can’t figure out what the unit of time is. Table 1 states 2024, but the time unit cannot be one year (2024) since I am reviewing this manuscript in 2024.
Answer: We agree. In order to avoid confusing, we changed “rate” to “prevalence” in Table 1.
5) Table 1: The prevalence/rate/percent of females with anti-HBc is 52.2, but line 141 uses the number 46.3%. I can’t tell if these are different units of measurement or this is a mistake. There are several other data points that seem discrepant between text and tables – please review all numbers carefully.
Answer: We apologise, this error has been corrected.
6) Line 150: Could there have been differences in late vaccination rates between males and females or other explanations for this difference?
Answer: We have added a paragraph to the Discussion “In this study, we found that the prevalence of HBsAg increases with age in males but not in females. This may be because males have more social interaction than females of an equivalent age, exposing them to infectious sources ofHBV.”
7) Line 209: Elsewhere the authors suggest that high titers of anti-HBs may be reflective of boosting from natural infection. Do the authors have previous anti-HBs titers to determine whether these people had low titers at some point in the past?
Answer: We found in our study that twenty-six subjects with anti-HBs negative in 2015 or 2019 become anti-HBs positive in 2024. Ten of them have anti-HBs titer of 1000IU/ml.
8) Line 235: This study demonstrates that we need to be more precise in our terminology. Birth vaccination provides strong (but not complete) protection from the development of chronic, active HBV infection (HBsAg positive) but protects less well from exposure (anti-HBc). The obvious question, which perhaps could be a future objective of this study, is whether detection of anti-HBc is associated with important health consequences (liver cancer, reactivation with immune suppression, etc). If that link is established, then the world may need to reconsider revaccinating people in adulthood.
Answer: We agree. And we have added a paragraph to recommend a future study to the Discussion.
9) Line 279: Those of us who practice in low-endemicity regions rarely see unusual combinations of HBV serologies such as HBsAg negative, HBeAg positive and HBV DNA negative. This study is important in demonstrating what is possible and what serology tests we should be following.
Answer: Thank you for appreciating this.
Comments on the Quality of English Language
This manuscript needs editing to improve clarity.
Answer: The manuscript has been edited by a native English speaker.
Reviewer 3 Report
Comments and Suggestions for Authors
Thank you for the opportunity to review manuscript ID: vaccines-3382993. This manuscript presents the prevalence of occult HBV infection in adults vaccinated against hepatitis B at birth.
The paper deals with a very important topic, which is an issue that has been of interest to all health professionals over the last several decades.
In the Introduction section, the importance of this issue around the world is presented in a high-quality and correct manner. The size of the problem of HBV infection, as well as the HBV-diseases-related burden, is presented in a logical order. In this same manner, the effect of HBV vaccine implementation on trends in HBV disease burden is shown too. In this context, the problem of occult HBV infection was highlighted. In the last paragraph of the Introduction section, the application of the plasma-derived HBV vaccine in the observed population is stated.
Comments:
- Indicate the vaccination coverage of the plasma-derived HBV vaccine in the described population as a whole.
- List data on the application of hepatitis B immunoglobulin being administered at birth, in those born to mothers who were positive for HBsAg in this population.
- In accordance with the logical order in which the information is presented in the Introduction section, add a paragraph in which data should be provided on whether the recombinant HBV vaccine was administered in this population, when the administration of the recombinant HBV vaccine began in the studied population, and what were the indications for the administration of this vaccines, what was the vaccination coverage.
- Specify the goal/goals of this work more precisely at the end of the Introduction section.
- In the Methods section, state `Study design`, `Study population`, `Study sample`, `Sample size calculation`, `Variables`, `Outcomes`.
- Lines 81-91: Add a new Figure where the `Flow chart of this study` will be displayed.
- Line 91: Specify inclusion and exclusion criteria. Specify the `Response rate`.
- Lines 89-90: Explain whether the questionnaire contained questions about personal health history, comorbidities, surgery or other medical procedures, blood transfusion in the population studied, etc., since it was stated, I quote `Each study subject completed a one-page questionnaire at that visit ...`.
- Line 305: The determinants of the occurrence of occult HBV infection in adults vaccinated against hepatitis B at birth, as well as the explanation of their mutual connection, have not been satisfactorily discussed. Correct this.
- The limitations of this study are not adequately stated in this paper. At the end of the Discussion section, add a new paragraph in which the limitations of this study should be listed and discussed, as well as the possibilities for overcoming them.
- Add a new paragraph on `Implications of this study`.
Author Response
Thank you for the opportunity to review manuscript ID: vaccines-3382993. This manuscript presents the prevalence of occult HBV infection in adults vaccinated against hepatitis B at birth.
The paper deals with a very important topic, which is an issue that has been of interest to all health professionals over the last several decades.
In the Introduction section, the importance of this issue around the world is presented in a high-quality and correct manner. The size of the problem of HBV infection, as well as the HBV-diseases-related burden, is presented in a logical order. In this same manner, the effect of HBV vaccine implementation on trends in HBV disease burden is shown too. In this context, the problem of occult HBV infection was highlighted. In the last paragraph of the Introduction section, the application of the plasma-derived HBV vaccine in the observed population is stated.
Comments:
- Indicate the vaccination coverage of the plasma-derived HBV vaccine in the described population as a whole.
Answer: We have pointed out this in the last paragraph of the Introduction section as “between 1986 and 1996, all newborn infants in these trials were vaccinated according to a 0-, 1-, and 6-month schedule using a 10 μg dose of plasma-derived HBV vaccine, regardless of the mother’s HBV infection status”
- List data on the application of hepatitis B immunoglobulin being administered at birth, in those born to mothers who were positive for HBsAg in this population.
Answer: We did not test the mothers’ HBV infection status. Therefore, none of newborn infants was injected with immunoglobulin.
- In accordance with the logical order in which the information is presented in the Introduction section, add a paragraph in which data should be provided on whether the recombinant HBV vaccine was administered in this population, when the administration of the recombinant HBV vaccine began in the studied population, and what were the indications for the administration of this vaccines, what was the vaccination coverage.
Answer: Agreed. We have added “All newborn infants in LongAn county were vaccinated with recombinant HBV vaccine from 1997.” As we pointed out in the Abstract, our study subjects recruited from those born between 1987-1993. Therefore, none of our study subjects was administered the recombinant HBV vaccine.
- Specify the goal/goals of this work more precisely at the end of the Introduction section.
Answer: Agreed. We have added “Does OBI prevail in this vaccinated cohort?” to that section.
- In the Methods section, state `Study design`, `Study population`, `Study sample`, `Sample size calculation`, `Variables`, `Outcomes`.
Answer: To organise our methods according to the suggested structure may not be better. In fact, all the information has been included in our paper. Furthermore, some other authors did not use the suggested structure (Zięba K, et al. Vaccines (Basel). 2024 Dec 29;13(1):18; Wang WC, Vaccines (Basel). 2025 Jan 20;13(1):95)
- Lines 81-91: Add a new Figure where the `Flow chart of this study` will be displayed.
Answer: Agreed and have done and added to the section of 2.1. Study Population.
- Line 91: Specify inclusion and exclusion criteria. Specify the `Response rate`.
Answer: Agreed. We have added the criteria below to the section of 2.1. Study Population
The inclusion criteria include: â‘ Born between 1987 and 1993. â‘¡ Came for our evaluation at least once. â‘¢ Had received three 10 μg doses of plasma-derived HB vaccine at ages 0, 1, and 6 months.
④ No history of blood transfusion. ⑤ No autoimmune liver disease, metabolic liver disease
or generalized metabolic disorder, etc. Those who had received the first dose of vaccine more than 72 h after birth were excluded.
The response rate was added to the section of 3.1. General Characteristics of the Study Subjects
- Lines 89-90: Explain whether the questionnaire contained questions about personal health history, comorbidities, surgery or other medical procedures, blood transfusion in the population studied, etc., since it was stated, I quote `Each study subject completed a one-page questionnaire at that visit ...`.
Answer: We have added the information below to section 2.1. Study Population. The questionnaire includes demographic information, including sex, birth date, ethnicity, place of birth, and immunization history. History of diseases such as autoimmune liver disease, metabolic liver disease
and generalized metabolic disorders, etc., and blood transfusion.
- Line 305: The determinants of the occurrence of occult HBV infection in adults vaccinated against hepatitis B at birth, as well as the explanation of their mutual connection, have not been satisfactorily discussed. Correct this.
Answer: Agreed. We have corrected this and added one more reference; please see that in Discussion.
- The limitations of this study are not adequately stated in this paper. At the end of the Discussion section, add a new paragraph in which the limitations of this study should be listed and discussed, as well as the possibilities for overcoming them.
Answer: Agreed and we have done this.
- Add a new paragraph on `Implications of this study`.
Answer: We think that the fifth paragraph should be enough to represent the implications of this study.
Round 2
Reviewer 2 Report
Comments and Suggestions for Authors
There is still some data confusion between the abstract (line 24), where female anti-HBc prevalence is 46.3% and line 146, where female anti-HBc prevalence is 52.2%. I think there may be a mix up between anti-HBs and anti-HBc prevalence? Please check all numbers in abstract, text and table.
Line 145 note anti-HBc, not HBc.
Author Response
There is still some data confusion between the abstract (line 24), where female anti-HBc prevalence is 46.3% and line 146, where female anti-HBc prevalence is 52.2%. I think there may be a mix up between anti-HBs and anti-HBc prevalence? Please check all numbers in abstract, text and table.
Answer: We apologise, this error has been corrected.
Line 145 note anti-HBc, not HBc.
Answer: We apologise, this error has been corrected.
Reviewer 3 Report
Comments and Suggestions for Authors
Thank you for the opportunity to re-review manuscript ID: vaccines-3382993.
The authors answered some questions, while they provided explanations for other questions. I thank the authors.